# Hindu Civilizationism: Make India Great Again

**Raja M. Ali Saleem** 

Independent Researcher, Islamabad 44000, Pakistan; alis141@gmail.com

**Abstract:** Hindu civilizationism is more than a century old phenomenon that has been steadily gaining strength. Its recent amalgam with populism has made it ascendant, popular, and mainstream in India. This paper explores how Hindu civilizationism is not only an essential part of the Hindutva and BJP's narrative but also the mainstay of several government policies. The "other" of the BJP's populist civilizationist rhetoric are primarily Muslims and Muslim civilization in India and the aim is to make India "vishwaguru" (world leader) again after 1200 years of colonialism. The evidence of this heady mixture of civilizationism and populism is numerous and ubiquitous. This paper analyzes topics such as Akhand Bharat, the golden age, denigrating Mughals, Hindutva pseudoscience, and Sanskrit promotion to highlight the evidence.

**Keywords:** India; Hindutva; populism; civilizationism; Vishwaguru; Modi; Sanskrit; Mughals; Akhand Bharat; RSS

## 1. Introduction

Civilizationism uses a religio-civilization classification of people to define national identity. Territorial nationalism is deemphasized as the nation is imagined beyond national boundaries. Citizens, who are considered part of the civilization-nation based on religion, are asked to defend or save their civilization which is considered under threat. The state becomes a means to achieve the objective which is civilizational longevity and success.

This civilizational rhetoric has become more common as populist leaders around the world have used it to attract voters who are dissatisfied with the dominant ideologies and established mainstream parties. Numerous authors have pointed out how rightwing populist European parties and leaders defined self and the other not in national but in broader civilizational terms. Christian civilization, Christian heritage, or Judeo-Christian civilization and traditions are considered in crisis and under threat from Islam and Muslims. The "patriots" are told they have only two choices, act or go extinct (Brubaker 2016, 2017; Kaya and Tecmen 2019; Ozzano and Bolzonar 2020; Yilmaz and Morieson 2021; Marchetti et al. 2022).

Populism is generally considered a thin ideology that attaches itself to rightwing or leftwing ideologies to give coherence, strength, and program to its rhetoric. Yilmaz and Morieson (2022a) have identified civilizationism as another thick ideology that populism attached itself to. Civilizationism, for them, is an idea that divides and categorizes people based on "civilizations" that are primarily based on religion. This is different from the usual division of people based on nations and populists' framing of "the people" as people of a country.

Hindu civilizationism, like in many other countries, is closely associated with rightwing nationalism and populism. Hindutva (literally Hindu-ness) is a popular political ideology that defines Indian values and nationalism primarily in terms of Hinduism and Hindu civilization, lays claim that only Hindus have the right to rule in India, and aims to replace a secular Indian constitution with a Hindu state (Hindu Rashtra). Hindutva political parties, organizations, and social movements raised the flag of civilizationism before Indian independence in 1947, and more than a century later, they are still its torchbearers. The ruling Bharatiya Janata Party (BJP) is just the latest and most successful of the political

rightwing conservative organizations. In terms of making Hindu civilizationism popular, the BJP plays second fiddle to almost a hundred-year-old Rashtriya Swayamsevak Sangh (RSS), a militant Hindutva organization.

Hindu civilizationism, therefore, is not a new phenomenon. It started as part of Hindu revivalism in the early 19th century. Hindu revivalists, such as Raja Ram Mohan Roy, the founder of Brahmo Samaj, wanted to reform Hinduism and Hindu society so that it could rise above the social evils and iniquitous rituals and regain its position as a great civilization. Later, in the last century, civilizationist organizations were formed whose main objectives were political even when they were not working as a political party. For a long time, from the 1940s to the 1970s, these organizations and the political parties they supported remained unpopular. However, civilizationists started winning elections at the state level in the 1980s and won national elections in the late 1990s. They ruled India under Prime Minister Vajpayee (1999–2004) but as they did not have the majority of Lok Sabha seats, they were always dependent on other parties and could not fully implement their agenda. The second decade of the 21st century brought a sea change in their fortunes as, like many rightwing civilizationists in other countries, they discovered an affinity between their ideology and populism. The embrace of populism under Prime Minister Modi made them the supreme political force and, currently, Hindu civilizationism is the dominant ideology in India. PM Narendra Modi has trounced the opposition in the 2014 and 2019 national elections and he is by far the most popular leader in India, most likely to win the 2024 national elections (Pradhan 2022).

During the last decade, Hindu civilizationism rhetoric has been rising steadily, with the support of populism. The people-elite divide of populism has been used to discredit the Congress Party and Nehru-Gandhi dynasty. Similarly, the people-outsider divide, another regular feature of populist politics, was used to declare Hindus as the only original inhabitants of India and Muslims and Christians as outsiders. Fear, threat, and crisis were used by Modi, like other populists, to force ordinary Hindus to be afraid (*Hindu khatray main hai*: Hindus are in danger) and act as advised by Modi. Finally, Modi's image as the only strong, decisive leader in India was carefully crafted, as in the case of many other populist leaders, to sway voters (*Modi hai to mumkin hai*: If there is Modi, then it is possible) (Sinha 2021; Saleem 2021; Saleem et al. 2022; Yilmaz and Morieson 2022b).

The othering of Muslims and other political forces as anti-nationals and non/fake Hindus is increasing in India. Hinduism has been presented as under threat from Muslims despite Hindus being close to eighty percent of the Indian population. Indians are being made to believe that Muslims and Westerners are again plotting to subjugate Hindus as they did many times during the previous twelve hundred years. The BJP has achieved what was unimaginable a few decades ago (Varshney 2019; Amarasingam et al. 2022).

However, it must be clear that Modi is not more civilizationist than previous civilizationists. The key difference between Modi and Hindu civilizationists of the 1960s and 1970s is the degree of populism, not the degree of civilizationism. Like Modi, Hindu civilizationists of the 20th century were also talking about Akhand Bharat, the golden Vedic age, oppressive Muslim/Mughal invaders, superiority of Hindu civilization, and Sanskrit promotion. Hindutva party manifestos of Hindu Mahasabha and Bharatiya Jana Sangh give ample evidence of civilizationalism (Saleem 2021). Populists are quite successful in the use of transnational solidarity today due to a number of reasons, such as revolution in information and communication technologies, globalization, increase in economic inequality and distrust in democratic institutions in numerous countries, rise in ethnic/religious attachments with the concurrent decline in liberalism, etc. However, they were not the first ones to use transnational solidarity. For instance, between 1987 and 1989, during the Ram Mandir movement, hundreds of thousands of bricks were donated for the eventual construction of Ram Mandir in Ayodhya. These bricks were donated in response to a campaign by the Vishwa Hindu Parishad (VHP), a Hindutva organization. The Ram Mandir movement and VHP campaign were pro-Hindutva and Hindu civilizationist. They were also

transnational as bricks for the mandir were donated not only by people living in India but also by people of 55 other countries of the world (Udayakumar 1997; Hindustan Times 2021).

## 2. Hindu Civilizationism

Like European civilizationism, Hindu civilizationism also builds on an identity that is not solely religious. From the start, both religion and ethnicity were used to explain Hindu civilizationism and to attract people to the cause, although Hindus do not belong to one ethnic group. Vinayak Damodar Savarkar, the first prominent ideologue of Hindu civilizationism, defined Hindutva in ethnic, political, and cultural terms and deemphasized the religious connection. The idea was to not reject anyone just because (s)he is from a different religion. Hindutva accepted other religions that it considered part of the Hindu civilization, although the three religions (Buddhism, Jainism, and Sikhism) that fulfill this criterion are not even three percent of the Indian population combined. For Savarkar, Hindus were a group of people for whom India was the motherland (*matrbhumi*), the land of ancestors (*pitrbhumi*), and the holy land (*punya bhumi*), a mixture of ethnicity and religion (Tharoor 2018, p. 249). For Savarkar, the idea of civilizationism was part of Hindutva, and Hindutva's followers' primary task was to save Hindu culture and civilization which was under threat from foreigners, primarily Muslims (Jaffrelot 2021, p. 13).

Madhav Sadashivrao Golwalkar, the second head of the RSS and one of the most important ideologues of Hindutva, in his book *We, or Our Nationhood Defined*, admired Nazi Germany's treatment of its minorities and clarified that Muslims (and Christians) have only two choices, assimilation or life without any rights: "[They] must either adopt the Hindu culture and language, must learn to respect and hold in reverence Hindu religion, must entertain no idea but those of the glorification of the Hindu race and culture . . . , or may stay in the country, wholly subordinated to the Hindu Nation, claiming nothing, deserving no privileges, far less any preferential treatment—not even citizen's rights." (Jaffrelot 2021, p. 14).

Like territorial nationalists, civilizationists also focus on history and create an imagined history based on myth making and invented traditions. Civilizationists portray and acclaim the greatness and magnificence of their civilization and compare how their civilization was better than other civilizations. The focus is on civilization, not nation, so the competition is also with other civilizations. As Yilmaz and Morieson (2022a) explain, populist civilizationist leaders do not limit themselves to applauding their civilization. There is also talk of danger and a crisis. The followers are informed that their civilization is in danger as adherents of other civilizations are trying to destroy it. The imminent threat discourse forces people to think in terms of civilizations, rather than nations, and binaries (us-them). In this zero-sum game, denigrating other civilizations is also necessary to keep them in a subordinate and inferior place.

In the following section, the prominence of Hindu civilizationism in present day India and Hindutva leaders' consistent efforts to prove the grandeur and glorious achievements of Hindu civilization will be shown. The section will also demonstrate how Hindu civilizationism is being promoted by denigrating Muslim civilization and its achievements. Most of the papers on civilizationism focus on Christian-majority and Muslim-majority countries. This paper analyzes civilizationism in India, a Hindu-majority country. The key theoretical contribution of this paper is that it presents the evidence that civilizationism as a concept is as applicable and functional in a Hindu majority country as it is in Christian, Jewish, and Muslim majority countries.

## 3. Golden Age

A golden age is a period in the history of a nation, group, or civilization during which not only was there peace and harmony but also great achievements in multiple fields, such as literature, politics, sports, science, etc., were also accomplished. For Greek civilization, the Classical period, the 5th to 4th century Before the Common Era, is considered the golden age when there was an end to tyranny and a time of great cultural growth. For Islamic civilization, the Abbasid Caliphate, from the 8th century to the 13th century is

considered the golden age. According to eminent Indian historian Romila Thapar (2002, p. 16), most 20th century Indian historians either participated in or were influenced by the independence movement so they tended to promote nationalist interpretations of history that applauded Indian history, culture, and traditions even when evidence was missing. The Indian tradition of non-violence was praised but so were Indian warrior kings. Democracy and constitutional monarchy were shown as part of Indian tradition, instead of colonial imports, and meetings of advisors of ancient Hindu kings (*mantriparishad*) were found comparable to the workings of the British Privy Council. Finally, different periods of ancient Indian history were depicted as golden ages to strengthen the claim of a great Hindu civilization:

> There was . . . an endorsement for the ancient past being a 'Golden Age'; such an age being a prerequisite for claims to civilization. This view was an inevitable adjunct to nationalist aspirations in the early twentieth century. The Golden Age was either the entire Hindu period that was seen as unchanging and universally prosperous, or else the reign of the Gupta kings which historians, both Indian and British, had associated with positive characteristics and revival of the Brahmanical religion and culture. (Thapar 2002, p. 17)

Anyone searching for the golden age of Indian history has to look no further than the (Muslim) Mughal Empire (1526–1857), especially the time of the Great Mughals (1526–1707). Evidence of Mughal greatness was everywhere in the form of magnificent buildings and architecture. There was also data to support it. During the times of the Great Mughals, there was unprecedented prosperity, and the Indian economy, along with the Chinese economy, dominated the whole world as almost one-fourth of the world's GDP came from India. By 1700, the Indian economy was the largest in the world in terms of GDP, surpassing the Chinese economy (Angus 2003, p. 261).

Shashi Tharoor (2016, p. 2) calls India the "glittering jewel of the medieval world" and quoted an American Unitarian Minister J. T. Sunderland to demonstrate what a colossus the Mughal Empire was before the conquest of India by the British. Writing in the early twentieth century, Sunderland praised the world-famous textile goods, jewelry, precious stones, pottery, porcelains, ceramics, businessmen, great architecture, goods made in iron, steel, silver, and gold, commerce, and engineering works of India. He wrote that India was not only ahead of others in producing magnificent traditional goods but also in industries and manufacturing:

> Nearly every kind of manufacture or product known to the civilized world—nearly every kind of creation of man's brain and hand, existing anywhere, and prized either for its utility or beauty—had long been produced in India. India was a far greater industrial and manufacturing nation than any in Europe or any other in Asia. (Sutherland 1929, p. 368)

Due to their hatred and rejection of Muslims, the Hindutva leaders and ideologues could not select the most obvious and pertinent choice, i.e., the Mughals. So, they had to not only find a new golden age but also destroy the Mughals' reputation and prestige. They did not have such a historically established golden age as in the cases of Greek, Islamic, or Egyptian civilizations, so they faced several problems. First, there is less and less evidence available as one moves back into Indian history. Archaeological evidence is limited and there is meager historical evidence that can be relied upon to establish facts. Even the archaeological evidence available was destroyed because of ineffective protection of various sites and population explosion (Allchin 1995, pp. 8–9). Myths and history are difficult to distinguish in religious texts that are available. George Erdosy (1995, p. 79) argues that the available texts of the period "are of primarily religious orientation, devoid of serious historical content." Second, undeterred by the lack of historical evidence, most Hindutva leaders choose the Vedic period, the period when Vedas, the earliest of Hinduism scriptures, were composed, which creates additional problems. Even if the Vedic period is accepted, the exact period of Vedas is not known. The rough estimates are between 1500

BCE and 500 BCE. Finally, many influential Hindutva ideologues, including Savarkar, do not accept the Vedic period as the only golden age. Savarkar (1971) wrote a book titled *Bharatiyil Itihaasil Saha Sone Pane* (*Six Glorious Epochs in Indian History*, 1971) in Marathi and identified not one but six glorious periods of Indian history. The first one of those glorious periods was the early Maurya Dynasty which existed from 322 to 260 BCE. However, it is the fifth glorious epoch that Savarkar focuses on. Marathas, because of their numerous successful battles against Muslims, were Savarkar's heroes. Others consider the age of Ram or Rama Chandra, the seventh avatar of Vishnu (one of the principal deities of Hinduism), as the glorious age of Hinduism. It is difficult to establish whether Ram existed or not, but some people have concluded that he existed around five thousand BCE, so much earlier than the Vedic period (ZeeNews Bureau 2012; Hindutva Watch 2019).

During the golden age of Indian/Hindu civilization, ancient India is presented by the Hindutva leaders as Vishwaguru (world leader), without much evidence. Sometimes, references are given to achievements of the (urban) Indus Valley Civilization, which is generally accepted as separate from the (rural) Vedic civilization (McIntosh 2008, p. 31). R. K. Pruthi (2004, pp. 237–42) gave detailed evidence that the Indus Valley civilization was not part of Vedic civilization, refuting the arguments given by Hindutva leaders and authors influenced by them.

In 2018, the then Vice President of India, M. Venkaiah Naidu, a former President of the BJP, without attribution or embarrassment, used Trump's slogan (Make America Great Again) as the title of his article, "Make India Viswaguru Again," to promote the idea that India was once the leader of the world and Indian civilization was the greatest (Naidu 2018).

The National Education Policy (NEP) 2020 was the first education policy formulated after the 1986 Education Policy. Its importance could not be exaggerated as the world of education has transformed during the thirty-four years between the two education policies. The NEP eulogizes the Hindu civilization and informs that ancient India had world-class educational institutions which produced great scholars. It is important to note that the focus of the NEP was primarily on non-Muslim institutions (such as Nalanda, Takshashila, Vallabhi, and Vikramshila) and scholars (such as Aryabhata, Bhaskaracharya, Brahmagupta, Chanakya, and Thiruvalluvar). The contributions of the Indo-Muslim civilization or Ganga-Jamuni civilization, the syncretic civilization based on the contributions of both Hindus and Muslims, were not highlighted (Government of India 2020).

## 4. Denigrating Mughals

Besides promoting ancient India as the golden age, the Hindutva leaders also denigrate the Indo-Islamic civilization in general and the Mughals in particular to support their anti-Muslim agenda. This was not true of early non-Hindutva leaders of India who generally praised the Mughals and appreciated the Mughal policy of tolerance and Sulah-e-Kul (peace to all). The following are a few excerpts from Nehru's book *The Discovery of India (1989)*, which praises the Muslim rulers of India for their support of Indian culture, music, poetry, literature, etc.:

> The record of the Indo-Afghan, Turkish, and Moghul rulers, apart from some brief puritanical periods, is one of definite encouragement of Indian culture, occasionally with variations and additions to it . . . It might be said that except in regard to actual image-making no attempt was made by Moslem rulers, apart from a few exceptions, to suppress any art-form. (Nehru 1989, p. 116)

Nehru was particularly appreciative of Akbar, the Great, and in his book, he included Akbar in the three personalities (along with Ashoka and Buddha) he admired (Nehru 1989, pp. 51–52). Nehru acknowledged that Akbar identified with India and was popular with Hindus and Muslims. He also declared the Mughals an Indian dynasty, facts vigorously denied by the Hindutva leaders (Nehru 1989, pp. 259–60).

Mahatma Gandhi also did not see the Mughal Era as a period of slavery or colonialism as present-day Hindutva present. He thought there was an element of local rule during the Mughal Era. In 1921, he said, "The pre-British period was not a period of slavery. We

had some sort of swaraj under the Mughal rule. In Akbar's time, the birth of a Pratap was possible, and in Aurangzeb's time, a Shivaji could flourish. Have 150 years of British rule produced any Pratap and Shivaji?" (Sahu 2022). In contrast to Nehru and Gandhi Ji, the Hindutva leaders are bent on erasing Muslim heritage in general and Mughals in particular from Indian history or relabeling and redesignating them as villains.

PM Modi and other Hindutva leaders regularly talk about "Bara so sal ki ghulami" (translation: twelve hundred years of servitude or slavery), thus depicting Muslim rulers (both rulers of Delhi Sultanate and Mughals) as colonizers like the British, even when only the first generation of the Mughals and other Muslim invaders were born outside India and had loyalties to areas outside India. The next generations, unlike British colonizers, were born in India and there is no evidence to conclude that they did not think of India as their only home. Mughals, unlike the British, were not robbing India and sending money to their homes thousands of miles away. As Nehru pointed out, Muslim invaders settled in India, adopted Indian cultural practices, married Indian women, and so they became as Indian as any other son of the soil.

In June 2014, in his very first address to the Lok Sabha after becoming the PM, Modi talked about twelve hundred years of slavery (Zee News 2014). A few months later, in his first trip to the US as PM, Modi again referred to the thousand or twelve hundred years of slavery "We are well aware of the history of our freedom movement. The Britishers ruled over us and prior to them various others ruled us. Almost for 1000 to 1200 years, we were slaves" (Government of India 2014). In 2019, PM Modi, while laying the foundation stone of a Mandir complex in Gujarat, again referred to the 1200 years of slavery: "It was due to our spiritual force that we were able to fight for our pride, our culture, and our traditions during the 1000–1200 years of our slavery." (PTI 2019).

As Audrey Truschke explains, this rewriting of history is being done to discredit, otherize, and exclude Muslims and to present the pre-Delhi Sultanate period as the only one that can be legitimately described as the golden age. She informs that "Hindutva history," an oxymoron, divides premodern India into two major phases, the Hindu golden age and Muslim colonialism. Unsurprisingly, Narendra Modi and other Hindutva leaders talk of "1200 years of slavery" (Truschke 2020).

Besides relabeling the Mughals as invaders and colonizers, the Hindutva civilizationists are also erasing signs of the Mughal era. In 2014, the BJP, soon after coming to power, changed the name of Aurangzeb Road to Abul Kalam Road in Delhi. In August 2018, Mughalsarai (meaning Mughal inn or tavern) Junction, an iconic railway station, was renamed by the BJP government after RSS ideologue and Bharatiya Jana Sangh President Deen Dayal Upadhyaya. In October 2018, the BJP government changed the name of Allahabad, a city founded by the Mughals, to Prayagraj. Again in 2018, Faizabad district was renamed Ayodhya district. Many people linked it to the 2019 national elections. More recently, in July 2022, the BJP coalition government in Maharashtra state approved the renaming of two cities. Aurangabad, named after Mughal Emperor Aurangzeb, will be renamed Chhatrapati Sambhaji Nagar, and Osmanabad, named after the Muslim ruler of Hyderabad state, shall be renamed Dharashiv (Sen 2019; Zee News 2022). More recently, the name of the iconic Mughal Gardens in Rashtrapati Bhavan, the official residence of the Indian President, was changed to Amrit Udyan (garden of the sacred nectar) in January 2023 (Kothari 2023). Soon afterwards, the "Mughal Garden" in the Delhi University's North Campus was renamed as the "Gautam Buddha Centenary" Garden. A university official claimed that it was a coincidence that the change of the university garden name came so close after the change of name of Rashtrapati Bhavan garden (Wire Staff 2023).

In addition to changing names, Hindutva leaders are also changing the curriculum to omit or downplay references to Mughals and other Muslim rulers. The focus has shifted from fact-based recent Indian history to ancient Indian history based on myths and religious texts. Back-to-back massive victories at the national level in 2014 and 2019 have allowed the BJP to cast a long shadow on the education system. Recently published textbooks promote Hindutva history and Vedic myths over fact-based history. Hindu rulers are

glorified, and Muslim rulers are vilified. For instance, a textbook declares five centuries of rule by Muslim kings, who were different and fought against each other, a single "Period of Struggle" and demonizes these rulers (Traub 2018). Furthermore, in a recent review of school textbooks, the National Council of Educational Research and Training removed sentences or paragraphs that challenged stereotypical fallacies about Muslims or mentioned some Mughal emperors and Jawaharlal Nehru (Muslim Mirror 2022).

## 5. Akhand Bharat

Another evidence of Hindu civilizationism and rejection of territorial nationalism is the concept of "Akhand Bharat" or undivided India. According to this concept, the whole Indian subcontinent, sometimes also including Myanmar and Southeast Asia, is one nation as they were once part of Hindu/Vedic civilization. This idea was embraced by the Hindutva leaders before the Partition in 1947 and, after the Partition, they blamed the Congress for allowing the Partition to happen and vowed to recreate Akhand Bharat. Savarkar, the RSS leadership during the last hundred years, and the leaders of Hindu Mahasabha, Bharatiya Jana Sangh, and BJP have all supported this idea.

In 1937, Savarkar reiterated during his presidential address at Hindu Mahasabha's annual session that "Hindusthan must remain one and indivisible" from Kashmir to Rameshwaram, and from Sindh to Assam (Sampath 2019). The president of Bharatiya Jan Sangh in the 1960s, Pandit DeenDayal Upadhyay, also supported the idea of Akhand Bharat. Pandit Upadhyay, who was also an RSS pracharak, said, "The word Akhand Bharat includes all those basic values of nationalism and an integral culture. These words include the feeling that this entire land from Attock to Cutack, Kutch to Kamrup and Kashmir to Kanya Kumari is not only sacred to us but is a part of us." (Organizer 2022).

The current leadership of India's ruling party, the BJP, also supports this notion of Akhand Bharat based on Hindu civilizationism. The General Secretary of the BJP Ram Madhav said in 2015 that Akhand Bharat would be created but there would be no war. Neighboring countries would join India through peaceful means and popular consent (Express News Service 2015). On 14 August 2022, Pakistan's independence day, the RSS chief Mohan Bhagwat reminded everyone that Akhand Bharat would happen but it would happen only when people talk and dream about it (PTI 2022).

The VHP celebrates Akhand Bharat Sankalp Diwas (Akhand Bharat Resolution Day) on 14 August, Pakistan's independence day, every year. In 2022, VHK Jammu and Kashmir held a program and besides the Indian subcontinent, they also included Afghanistan, Tibet, and Thailand:

> Speaking on the occasion, speakers said that Akhand Bharat means bringing those areas of India back which were its part and parcel in ancient times. They said Afghanistan, Pakistan, Bangladesh, Sri Lanka, Burma, Malaysia, Tibet, Thailand, and other countries were part of United India. (Daily Excelsior 2022)

In prehistoric times, Hindu civilization influenced Southeast Asia, including Thailand, Cambodia, Indonesia, Laos, and Malaysia. The influences of Hindu civilization are still visible in these countries. The above quote makes it clear that the idea of Akhand Bharat is based on Hindu civilizationism as, otherwise, it will be close to impossible to explain how Malaysia or Thailand and India can or should become one country (Sengupta 2017).

## 6. Promotion of Sanskrit

Linked with the superiority of Vedic/Hindu civilization is the belief that Sanskrit, the language of Vedas and other Hindu religious texts, is superior to and/or the mother of all languages, and speaking it has several advantages (Express Web Desk 2019). It is called Dev Basha or the language of gods. It is also claimed that science, mathematics, and philosophy have their roots in Sanskrit, and speaking Sanskrit is a sign of a cultured society (ANI 2022). The imagined superiority of Sanskrit further strengthens the claim of a magnificent ancient Hindu civilization.

The Indian love affair with Sanskrit is old. In 1951, when the Indian Constitution was promulgated, Sanskrit was only one of the fourteen (14) scheduled languages (as they were part of the Eighth Schedule of the Indian Constitution) that was not spoken by anyone as a mother tongue. Now, there are twenty-two (22) languages in the Eighth schedule. This means that the founding fathers of India preferred Sanskrit over eight languages that were spoken by millions of Indians as their mother tongue. Article 351 also confers special status to the Sanskrit language. It directs the government to promote the Hindi language (which was designated as the official language by Article 343) and to enrich its vocabulary primarily using the Sanskrit language (Ministry of Home Affairs 2023). In 1969, it was the Congress government of Mrs. Indra Gandhi that decided to celebrate Sanskrit Diwas or Sanskrit (commemoration or celebration) day at the national level (Goswami 2019). However, present-day Hindutva leaders have praised and celebrated Sanskrit at a whole other level. They are not only trying to enrich Hindi with Sanskrit vocabulary but are also trying to revive the Sanskrit level as explained in the next paragraphs.

PM Modi regularly celebrates Sanskrit and promotes its revival. In 2012, while honoring a Sanskrit scholar, he told the gathering that he has established a Sanskrit university in Gujarat and his government has worked to teach one hundred thousand families (around eight hundred thousand people) to speak Sanskrit. He also claimed that in Germany, programs in Sanskrit were being broadcasted on the radio before the same was done in India (Modi 2012). In 2015, he was the first Prime Minister to speak many lines from Hindu sacred texts in Sanskrit during his speech to the UN General Assembly (Modi 2015).

In 2021, while talking to the nation in *Mann ki Baat*, he encouraged learning of Sanskrit as according to him it nurtured knowledge as well as national unity and good relations with other countries. He declared that Sanskrit literature is the divine philosophy of humanity and knowledge (TNN Bureau 2021).

The National Education Policy 2020 also confers special status to Sanskrit and the policy proclaims that Sanskrit will be mainstreamed. No other language, not even Hindi, which is another favorite of the Hindutva leaders, has been praised as much as Sanskrit in the NEP. The NEP declares that Sanskrit will be made available to students all over India in several ways. First, Sanskrit will be included as an option in the three-language formula. Second, Sanskrit will be made available as an optional subject at all education levels of school and higher education. Third, Sanskrit universities will not only teach Sanskrit and related subjects but will offer all major subjects and will become large multidisciplinary institutions of higher learning. Other universities will offer the Sanskrit language as a subject. Finally, Sanskrit and other classical language institutes and departments across the country will be strengthened. Ancient manuscripts will be preserved, and training will be provided to many students to study the manuscripts and their interrelations with other subjects (Teachers Adda247 2020).

## 7. Hindutva Pseudo Science

Hindu civilizationists believe that Hindus, Vedic culture, and Hinduism are the best and India was Vishwa guru before Muslim and Western colonialists invaded them. Therefore, it is not enough to show Hinduism's superiority over Muslim or Islamic civilization; they must compete with the best, the Western civilization. Since science and technology are the areas associated with the most prominent accomplishments of the West, the Hindu civilizationists show they are better than the West by claiming that the scientific advancements of today were achieved by Vedic India thousands of years ago.

Meera Nanda (2016) argues that Hindutva believers experience cognitive dissonance as the reality seems to contradict their belief that Hindu or Vedic civilization was the best. They experience a mixture of ressentiment and envy.

> The problem is this: We [Hindu nationalists] can neither live without modern science and the technologies it has spawned, nor can we make peace with the fact that this most fertile and powerful of all knowledge traditions is, after all, a *melechha* tradition. It rankles with us that these impure, beef-eating "material-



ists" . . . managed to beat the best of us when it came to nature-knowledge . . . . (Nanda 2016)

The melechha (foreign or barbarian in Sanskrit) civilization cannot be superior to the Vedic civilization and this has been propagated not only by the Hindutva leaders but also by Hindu revivalists since Swami Vivekananda and Dayananda Saraswati. Vedic/Hindu civilization being the greatest of all civilizations and the mother of all that is good in other civilizations was promoted as a fact by most revivalists/nationalists, and so it was easy to also claim that "Vedas are the mother of science." Swami Vivekananda claimed that Hinduism (Advaita Vedanta school) is closest to science and appeals to modern scientists (Tharoor 2018, p. 220). To prove this theory of Hinduism being the most "scientific" religion, ancient rishis (Hindu saints or sages) are presented as scientists who knew (and used) rules of genetics, physics, mathematics, avionics, etc., thousands of years before the Western scientists "discovered" it.

Some of the evidence presented to support the superiority of Hindu civilization borders rejection not only of history but also of rationality and reality. In its 2009 General Election manifesto, the BJP claimed that rice yields in India stood at 20 tons per hectare in ancient times, twice the yields today with all the modern science and technology (Tharoor 2018, p. 371). In 2012, when PM Modi was the Chief Minister of Gujarat, he falsely claimed that children in London were being taught Vedic Mathematics (Modi 2012). In 2014, PM Modi claimed, while addressing doctors and health professionals, that knowledge of cosmetic surgery and genetics was available in India thousands of years ago. He presented the examples of the Hindu god Ganesh and warrior Karna from Mahabharata to prove his point (Rahman 2014). In 2018, India's junior education Minister, Satyapal Singh, an MPhil in Chemistry, twice rejected Darwin's theory of evolution as he did not consider himself a "child of monkeys" and sought to remove it from the curriculum (Scroll 2018). Also in 2018, Biplab Deb, the BJP Tripura Chief Minister, claimed that ancient India had internet and satellite communications (Sanyal 2018). Finally, in 2021, during the COVID-19 pandemic, the health minister of India, Dr. Harsh Bardhan, helped launch the Coronil tablet of Patanjali, a company that develops ayurvedic medicine, as a much better cure for the deadly virus as compared to the Western medicine (The Hindu 2021).

These are just a few of the pseudo claims made by people working at the highest level in the Government of India. There are many more, including the claim that Stephen Hawking said that the Vedic theories were superior to Einstein's renowned equation, $E = MC^2$. Almost all of these claims were made by people close to the BJP. The earnest desire to prove Vedic knowledge superior or equal to modern science is dangerous for the new generations of Indians as it promotes fake history and fake science. But such bogus claims are regularly being made because it appears people who are now part of the government, including PM Modi, believe in the (fake) superiority of Vedic "science" and Vedic civilization. Therefore, such claims are readily accepted and even encouraged by the government, which is disastrous for India's future. Even the highest scientific organizations are not spared. Gauhar Raza, former chief scientist at the Council of Scientific and Industrial Research (CSIR), an organization constituting of around 40 national labs, has warned:

> Modi has initiated what may be called 'Project Assault on Scientific Rationality. A religio-mythical culture is being propagated in the country's scientific institutions aggressively." (Kumar 2019)

## 8. Conclusions

Globally, civilizational populism has become more popular during the last few decades. Mostly, civilizational arguments are the mainstay of rightwing nationalists. In almost all advanced democracies of the North as well as fragile democracies of the South, one sees right wing nationalists make civilizational arguments. Hindu civilizational populists are ahead of other civilizational populists as they have a strong leader who has captured the imagination of the voters and the support of the RSS that has been working on the Hindu civilizationist agenda for more than a century (Yilmaz and Morieson 2023, pp. 1–44, 181–23).

Hindu civilizationism started in the nineteenth century as Hindu revivalists tried to inspire their community to take pride in who they were and forge ahead. In the twentieth century, Hindu civilizationism's banner was kept afloat by Hindutva leaders. The Indian government, led by the Congress Party, was more interested in promoting an Indian civilization to which all religious and ethnic communities contributed. The focus was on science, development, and the future, not on the past, especially not the mythological past. PM Nehru was the epitome of this modernist attitude that remained supreme for the first fifty years of the Indian state. The Hindutva leadership and their civilizationism remained on the fringes. The civilizational arguments gradually grew in significance as Hindutva ideology became more popular. The 1990 Advani's Ram Rath Yatra was the biggest mobilizing event in the history and it brought Ram, Ayodhya, and Hindu civilization to the center of Indian politics (Frontline 2022). In 1998, Hindutva came to power at the federal level for the first time and, after 2014, it became the dominant ideology as Modi used his populist message to enamor a large section of Indian population. As explained above, Hindu civilizationism is Modi government's policy. Eulogizing the Vedic golden age, mainstreaming Sanskrit, yearning for Akhund Bharat, denigrating Mughals, and supporting pseudo (Vedic) science is now regularly done by the federal and state ministers of the BJP governments. The Hindutva nationalism is civilizational, not territorial, and the combination of Hindu civilizationism and populism has proven to be a winning formula for the BJP during the last decade.

The future of Hindu civilizationism appears bright in India. PM Modi is the most popular politician in India at the moment and appears likely to win a third term. Congress, the main opposition party, has elected a new president after more than three years, and Rahul Gandhi's Bharat Joro Yatra (journey/pilgrimage to unite Bharat/India) is drawing large crowds along thousands of miles of its route. It certainly presents Rahul Gandhi, the heir of the famous Nehru-Gandhi dynasty, as a popular politician that can give PM Modi a tough contest, but with less than eighteen months remaining in the 2024 general election, the opposition has a steep road to climb (Banerjee 2023; Biswas 2022). The question has also been asked about whether Gandhi is rejecting Hindu civilizationism outright and trying to create a secular state as envisioned in the Indian Constitution or wants a soft Hindu civilizationism of the former BJP PM Vajpayee brand that gives minorities, especially Muslims, some space to prosper (Barman 2022). Visiting Vajpayee's memorial in New Delhi during his yatra and naming his rally a "yatra", a term generally used for pilgrimage to Hindu religious sites, suggests that Gandhi still is not clear on which way he wants to go. This means that Hindu civilizationism may not be in danger even if the opposition wins in the 2024 general election.

**Funding:** This research received no external funding.

**Institutional Review Board Statement:** Not applicable.

**Informed Consent Statement:** Not applicable.

**Data Availability Statement:** Not applicable.

**Conflicts of Interest:** The author declares no conflict of interest.

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
