# Peer review of "Hindu Civilizationism: Make India Great Again"

_religions, doi:10.3390/rel14030338_

Round 1

Reviewer 1 Report

Review comments on Religions article – Hindu populism

General comments

This under-developed paper is long on outrage and emotive claims but well short on scholarly academic discourse. With perhaps a quarter of all words at present in the form of long direct quotes (unexplained for the most part), the author does not have much to say in their own words except to keep bemoaning the dire aspects of Hindu civilisationalism in contemporary India. In my view, the content of this paper does not represent anything new in the field. The prose is often less than satisfactory because there are so many unsupported claims and generalisations.

Specific comments

p. 1 The large direct quote from Yilmaz and Morieson (2022) needs a page number for the reference. In my view the quote is too long anyway. Most of it should be paraphrased to offer a succinct but properly referenced definition, and then explained in your own words for the reader.

p. 2. Line 74, ‘otherization’ is an awkward word. I would prefer the more usual ‘othering’.

p. 2. Lines 54-83 contains some major (and sensationalist) claims about the current Indian Prime Minister’s actions/intentions that are not supported by appropriate references. It reads as opinion, not academic discourse.

p. 2/3 The Tharoor 2018 direct quote than runs from line 94 to line 104 is much too long. Please reduce it to less than 40 words and paraphrase other points in the quote in your own words. The subsequent quotes from Jaffrelot 2021 are both also too long and I think unnecessary to include as direct quotes when the points could be better paraphrased and/or explained in your own words, with the specific references included of course.

p. 4 line 155. Since not all readers of this journal will be familiar with the history of India it should be pointed out at the beginning of this paragraph that the Mughal Empire was a Muslim Empire. 

p. 4. Line 166-179. The direct quote from Tharoor 2016 is much, much too long. It should be less than 40 words, with accompanying explanation by the author of the paper.

p. 5. The two extensive quotes on this page are once again much too lengthy and are included without adequate explanation. Both must be reduced to less than 40 words and should be embedded in commentary and explanation by the author of the paper that anchors them to the evolving academic argument.

p. 6. The two extensive quotes on this page are once again much too lengthy and are included without adequate explanation. Both must be reduced to less than 40 words and should be embedded in commentary and explanation by the author of the paper that anchors them to the evolving academic argument such as it is.

p. 6. Line 290-291 the claim that ‘PM Modi and other Hindutva leaders regularly talk about “Bara so sal ki ghulami” (translation: Twelve hundred years of servitude or slavery)’ needs a reference or references which support this claim.

p. 7. The long quotes from Trushke 2020 and Traub 2018 need to be reduced to less than 40 words and explained for the reader. Also, both citations need page numbers.

p. 8. The three direct quotes on this page are once again too long. Perhaps they could be cut and the points made in the author’s own words. If included then adequate explanation for the reader is needed. Each must be reduced to less than 40 words.

p. 8 lines 390-391 This is a complex point of argument and requires more than a single sentence to conclude the sub-section of discussion.

p. 8/9 lines 393-412 There are many substantial and controversial claims made in these two paragraphs. However, there are no supporting references. So it does not read as proper scholarly prose.

p. 9. The long quote attributed to PM Modi and the even longer quote from Nanda 2016 (which continues onto page 10) both need to be reduced to less than 40 words and explained for the reader. Page numbers should be included where possible.

p. 10. The ‘list’ of ‘evidence’ on this page is messy and confusing for the reader. It is also not a scholarly way of presenting sources to advance a particular argument in an academic paper. The direct quote by Rahman embedded in the middle of the list is too long and out of place. The brief discussion that follows the ‘list’ (and carries over to page 11) is full of emotive claims and unsupported assertions.

p. 11 the direct quote from Kumar 2019 is too long and is not explained for the reader.

p. 11. The two paragraph conclusion is slight and uninteresting. It repeats more or less material presented at the start of the paper and ends inconclusively.

Author Response

I have reviewed the paper in view of the respected reviewer's comments and did not find any outrage or emotive claims. In fact, one of the other reviewers wrote, "I found the manuscript rich in empirical terms." So, I focused on specific comments.

Specific comments

1. P.1 The large direct quote from Yilmaz and Morieson (2022) needs a page number for the reference. In my view the quote is too long anyway. Most of it should be paraphrased to offer a succinct but properly referenced definition, and then explained in your own words for the reader.

Done. The large direct quote was removed and paraphrased in my own words as suggested by the reviewer. The direct quote was removed and the reference is from a journal article so the page number is not required.

 2. P.2 Line 74, ‘otherization’ is an awkward word. I would prefer the more usual ‘othering’.

Done. ‘Otherization’ changed to ‘othering’ as suggested by the reviewer.

3. . P. 2 Lines 54-83 contains some major (and sensationalist) claims about the current Indian Prime Minister’s actions/intentions that are not supported by appropriate references. It reads as opinion, not academic discourse.

Done. PM Modi's name was removed. References added.

4. 2/3 The Tharoor 2018 direct quote than runs from line 94 to line 104 is much too long. Please reduce it to less than 40 words and paraphrase other points in the quote in your own words. The subsequent quotes from Jaffrelot 2021 are both also too long and I think unnecessary to include as direct quotes when the points could be better paraphrased and/or explained in your own words, with the specific references included of course.

Done. Direct quotes from Tharoor and Jaffrelot were both removed and their ideas were paraphrased as suggested by the reviewer.

5. p.4 line 155. Since not all readers of this journal will be familiar with the history of India it should be pointed out at the beginning of this paragraph that the Mughal Empire was a Muslim Empire. 

Done. Muslim was added before Mughals.

6. Line 166-179. The direct quote from Tharoor 2016 is much, much too long. It should be less than 40 words, with accompanying explanation by the author of the paper.

Done. The direct quote from Tharoor was shortened and his ideas were paraphrased as suggested by the reviewer.

7. p.5 The two extensive quotes on this page are once again much too lengthy and are included without adequate explanation. Both must be reduced to less than 40 words and should be embedded in commentary and explanation by the author of the paper that anchors them to the evolving academic argument.

Done. Direct quotes from VP Naidu and the NEP were removed and ideas were paraphrased and embedded as suggested by the reviewer.

8. p.5 The two extensive quotes on this page are once again much too lengthy and are included without adequate explanation. Both must be reduced to less than 40 words and should be embedded in commentary and explanation by the author of the paper that anchors them to the evolving academic argument such as it is.

Done. One direct quote from Nehru was removed and the other was shortened and ideas were paraphrased and embedded as suggested by the reviewer.

p. 6. Line 290-291 the claim that ‘PM Modi and other Hindutva leaders regularly talk about “Bara so sal ki ghulami” (translation: Twelve hundred years of servitude or slavery)’ needs a reference or references which support this claim.

Please see lines 300-310 where I have given three references of PM Modi's speeches, including his actual words (translated).

p. 7. The long quotes from Trushke 2020 and Traub 2018 need to be reduced to less than 40 words and explained for the reader. Also, both citations need page numbers.

Done. Both direct quotes were removed and ideas were paraphrased and embedded as suggested by the reviewer.

p. 8. The three direct quotes on this page are once again too long. Perhaps they could be cut and the points made in the author’s own words. If included then adequate explanation for the reader is needed. Each must be reduced to less than 40 words.

Done. Two direct quotes were removed, the third was shortened, and ideas were paraphrased and embedded as suggested by the reviewer.

p.8 lines 390-391 This is a complex point of argument and requires more than a single sentence to conclude the sub-section of discussion.

Done. The point is explained as suggested by the reviewer and a reference is also added.

P. 8/9 lines 393-412 There are many substantial and controversial claims made in these two paragraphs. However, there are no supporting references. So it does not read as proper scholarly prose.

Done. References are added.

P. 9. The long quote attributed to PM Modi and the even longer quote from Nanda 2016 (which continues onto page 10) both need to be reduced to less than 40 words and explained for the reader. Page numbers should be included where possible.

Done. The direct quote from Modi was removed and Nanda's quote was shortened and ideas were paraphrased and embedded as suggested by the reviewer.

P. 10. The ‘list’ of ‘evidence’ on this page is messy and confusing for the reader. It is also not a scholarly way of presenting sources to advance a particular argument in an academic paper. The direct quote by Rahman embedded in the middle of the list is too long and out of place. The brief discussion that follows the ‘list’ (and carries over to page 11) is full of emotive claims and unsupported assertions.

Done. The ‘list’ of ‘evidence’ (in bullet points) was changed into a paragraph and the direct quote by Rahman was removed.

The brief discussion that follows the 'list' is based on the evidence given above. It fails me how it is based on unsupported assertions.

P. 11 the direct quote from Kumar 2019 is too long and is not explained for the reader.

Done. Kumar's quote is shortened and further explained.

P. 11. The two paragraph conclusion is slight and uninteresting. It repeats more or less material presented at the start of the paper and ends inconclusively.

The conclusion has been expanded. 

Reviewer 2 Report

Review of the paper

“Hindu civilizationism: Make India Great Again”

Journal: Religions

This manuscript aims to explore the role of Hindu civilizationism in BJP's narrative. To this end, the author(s) address(es) different expressions of the “us” and the “other” in BJP's populist civilizationist rhetoric, while identifying different topics of analysis: the denigration of Mughals, pseudoscience based on Hindutva, and the promotion of Sanskrit.

I found the manuscript rich in empirical terms; nevertheless, the manuscript still has some theoretical gaps that require major revisions. This includes better addressing the theoretical mixture of civilizationism and populism, connecting these ethnonational and ethnoreligious components with the transnational dimension of contemporary populism, engaging with the emergent academic discussions on the gap between populism and foreign policy, and referring to similar processes in other places of the Global South, like Latin America or the Middle East.

I tend to think that, after these major revisions, the piece may and should be publishable. I hope to contribute through my comments to increase the chances of a revised manuscript, which can contribute to a greater understanding of the role of ethnoreligious and ethnonational elements in populism in general, and in International Relations in particular.

Specific comments:

-          Indeed, when referring to the theoretical framework of “civilizational populism”, the manuscript overlooks existing research on the populists’ strategic use of ethnonational and ethnoreligious components for (de)legitimation, thus influencing their foreign policy. Although there are some differences with the referred case, some lines of continuity can be recognized, including with reference to the need to balance between the populist civilizational discourse and the country’s geopolitical needs. I strongly recommend the author(s) to further emphasize the International Relations dimensions of the case, by exploring the emerging literature that focuses on the role of transnational features in Populism in general, and Populist Foreign Policy in particular.

-          In a similar vein, it can be claimed that also several non-populist parties use ethnonational and ethnoreligious discourse such as the BJP does. So, what makes populists “more” civilizational than non-populists? What is so special about populist leaders projecting these transnational constructions “abroad”? Populist ‘civilizationalism’ should be distinguished from similar non-populist strategies on some ontological basis – in other words, why populists use this type of transnational solidarity between “Peoples” for “legitimizing” themselves or “delegitimizing” their opponents? The author(s) can further elaborate on these points while connecting it with approaches to populists’ performances in the international scene.

-          Moreover, the author(s) is/are encouraged to identify similarities and differences with similar ideational constructions in the Global South, when referring to experiences of “Making XXX Great Again” that crosses national borders. As they stand, the rich empirics are rather skimpy to achieve a more complex analysis of civilizational populism in the Global South, and the literature review seems quite scant and superficial. The author(s) could make the case by referring to the existent evidence gathered about populists’ (de-)legitimation strategies in other areas, and thus gaining by contextualizing the phenomena of populist civilizational discourse in the Global South more broadly. For instance, BJP’s narrative shows similarities and differences with the third wave of populism in Latin America, which also drew on these ethnonational narratives to legitimize their own process of accumulation of power in their countries, their region, and abroad, and delegitimizing external opponents (Venezuela’s Chavismo in particular, but also Bolivia’s Evo Morales, Nicaragua’s Daniel Ortega, and Ecuador’s Rafael Correa, who addressed “Nuestramerica” as clashing with other extra-civilizational powers). I would recommend discussing, before the case study, whether the analysed case is different from previous cases/waves of populism and drawing on this to explain what is the core of civilizational populism today.

-          I tend to believe that refining these points could help develop the paper further and, in this way, contributing more substantially to the literature on global populism, civilizational populism, and populist foreign policy in the Global South.

Author Response

I found the manuscript rich in empirical terms; nevertheless, the manuscript still has some theoretical gaps that require major revisions. This includes better addressing the theoretical mixture of civilizationism and populism, connecting these ethnonational and ethnoreligious components with the transnational dimension of contemporary populism, engaging with the emergent academic discussions on the gap between populism and foreign policy, and referring to similar processes in other places of the Global South, like Latin America or the Middle East.

I tend to think that, after these major revisions, the piece may and should be publishable. I hope to contribute through my comments to increase the chances of a revised manuscript, which can contribute to a greater understanding of the role of ethnoreligious and ethnonational elements in populism in general, and in International Relations in particular.

Response: I have tried to address the relationship between civilizationism and populism but could not discuss the important international relations and foreign policy dimensions as they were not the focus of my article. However, I take note of this gap and hopefully write an article prioritizing on these dimensions.

Specific comments:

  1. Indeed, when referring to the theoretical framework of “civilizational populism”, the manuscript overlooks existing research on the populists’ strategic use of ethnonational and ethnoreligious components for (de)legitimation, thus influencing their foreign policy. Although there are some differences with the referred case, some lines of continuity can be recognized, including with reference to the need to balance between the populist civilizational discourse and the country’s geopolitical needs. I strongly recommend the author(s) to further emphasize the International Relations dimensions of the case, by exploring the emerging literature that focuses on the role of transnational features in Populism in general, and Populist Foreign Policy in particular.

Response: I could not discuss the important international relations and foreign policy dimensions of populism as they were not the focus of my article. However, I take note of this gap and hopefully write an article prioritizing these dimensions.

  1. In a similar vein, it can be claimed that also several non-populist parties use ethnonational and ethnoreligious discourse such as the BJP does. So, what makes populists “more” civilizational than non-populists? What is so special about populist leaders projecting these transnational constructions “abroad”? Populist ‘civilizationalism’ should be distinguished from similar non-populist strategies on some ontological basis – in other words, why populists use this type of transnational solidarity between “Peoples” for “legitimizing” themselves or “delegitimizing” their opponents? The author(s) can further elaborate on these points while connecting it with approaches to populists’ performances in the international scene.

Response: Please see lines 81-99 where I answer questions such as what makes populists “more” civilizational than non-populists and what is so special about populist leaders projecting these transnational constructions “abroad”.

Again, I would love to study and write about “approaches to populists’ performances in the international scene.” However, this is not the focus of this article.

  1. Moreover, the author(s) is/are encouraged to identify similarities and differences with similar ideational constructions in the Global South, when referring to experiences of “Making XXX Great Again” that crosses national borders. As they stand, the rich empirics are rather skimpy to achieve a more complex analysis of civilizational populism in the Global South, and the literature review seems quite scant and superficial. The author(s) could make the case by referring to the existent evidence gathered about populists’ (de-)legitimation strategies in other areas, and thus gaining by contextualizing the phenomena of populist civilizational discourse in the Global South more broadly. For instance, BJP’s narrative shows similarities and differences with the third wave of populism in Latin America, which also drew on these ethnonational narratives to legitimize their own process of accumulation of power in their countries, their region, and abroad, and delegitimizing external opponents (Venezuela’s Chavismo in particular, but also Bolivia’s Evo Morales, Nicaragua’s Daniel Ortega, and Ecuador’s Rafael Correa, who addressed “Nuestramerica” as clashing with other extra-civilizational powers). I would recommend discussing, before the case study, whether the analysed case is different from previous cases/waves of populism and drawing on this to explain what is the core of civilizational populism today.

Response: There are no doubt similarities between populism in different countries of the Global South. However, this is not the focus of this article. I have written extensively on civilizationism and populism in Pakistan and have recently submitted an article on civilizationism in Israel. However, it is respectfully submitted that this is not the focus of this article.

  1. I tend to believe that refining these points could help develop the paper further and, in this way, contributing more substantially to the literature on global populism, civilizational populism, and populist foreign policy in the Global South.

Response: Thanks, sir.

Reviewer 3 Report

No direct comments for the author.

Author Response

There are no reviewer comments to respond to.

Reviewer 4 Report

The most original contribution of this article is its argument that Hindu nationalists, since achieving national power in 2014, have asserted Hindu civilizational superiority in similar ways to other right wing nationalists and populists. Since the article does not draw on primary sources, it must be evaluated on the basis of its conceptual contributions. I have several questions/suggestions which would strengthen the analysis:

1.     Clarity about the nature of civilizational arguments. Do all right wing nationalists make civilizational arguments? Where would the author locate Hindu nationalists in relation to other right wing nationalists?

2.     The temporal dimension: as the author points out, some of the seminal Hindu nationalist ideologues were claiming Hindu civilizational superiority in the early 20th century. How/why have civilizational arguments become more significant since the BJP was elected in 2014?

3.     Populism: the author describes the current BJP government as populist but doesn’t explain how and why it’s populist or how populism buttresses the government’s claims of civilizational superiority. See for example this sentence, “Hindu civilizationism, like in many other countries, is closely associated with (46) rightwing nationalism and populism.”

4.     The paper states that this argument…”will also demonstrate how Hindu civiliza- (135) tionism is being promoted by denigrating other civilizations (primarily Muslim but also 136 (Christian) and their achievements. “Although they do an excellent job in demonstrating the denigration of Islam, they could better document the denigration of Christianity. More to the point, my own view is that Hindu nationalists’ views of Western civilization is filled with ambivalence which results both from their geo-political ambitions and the complex legacies of colonialism.

Author Response

The most original contribution of this article is its argument that Hindu nationalists, since achieving national power in 2014, have asserted Hindu civilizational superiority in similar ways to other right wing nationalists and populists. Since the article does not draw on primary sources, it must be evaluated on the basis of its conceptual contributions. I have several questions/suggestions which would strengthen the analysis:

  1. Clarity about the nature of civilizational arguments. Do all right wing nationalists make civilizational arguments? Where would the author locate Hindu nationalists in relation to other right wing nationalists?

Response: Please see lines 478-488. It answers the above questions. A detailed answer would require two to three pages which will make this article lose its focus.

  1. The temporal dimension: as the author points out, some of the seminal Hindu nationalist ideologues were claiming Hindu civilizational superiority in the early 20thcentury. How/why have civilizational arguments become more significant since the BJP was elected in 2014?

Response: Please see lines 48-99. As explained, civilizational arguments have become more significant because of a number of reasons and this led to the BJP success in the 1990s. Since 2011-12, Modi’s populism and popularity gave further impetus to civilizational arguments. Finally, after 2014, the BJP controlled state resources which were used to make civilizational arguments more significant and more prevalent as we see under the headings of denigrating Mughals, Hindutva pseudo science, promotion of Sanskrit, Akhand Bharat, etc.

  1. Populism: the author describes the current BJP government as populist but doesn’t explain how and why it’s populist or how populism buttresses the government’s claims of civilizational superiority. See for example this sentence, “Hindu civilizationism, like in many other countries, is closely associated with (46) rightwing nationalism and populism.”

Response: Please see lines 66-75. This paragraph explains how populism is used to help and support civilizationism.

  1. The paper states that this argument…”will also demonstrate how Hindu civiliza- (135) tionism is being promoted by denigrating other civilizations (primarily Muslim but also 136 (Christian) and their achievements. “Although they do an excellent job in demonstrating the denigration of Islam, they could better document the denigration of Christianity. More to the point, my own view is that Hindu nationalists’ views of Western civilization is filled with ambivalence which results both from their geo-political ambitions and the complex legacies of colonialism.

Response: Reference to the Christian civilization has been removed. Now, the sentence is, “will also demonstrate how Hindu civilizationism is being promoted by denigrating Muslim civilization and its achievements.” I agree with the respected reviewer's viewpoint that “Hindu nationalists’ views of Western civilization is filled with ambivalence.”

Round 2

Reviewer 1 Report

The paper is much improved. All the new material draws the reader's attention to what this essay contributes to what we know about civilisational populism. The addition of specific references has much improved the orientation of the paper.

Author Response

Thank you, reviewer 1. There are no specific changes required.

Regards.

Reviewer 4 Report

The author added some new material to the manuscript in response to two of the three questions I raised (and felt that they had already addressed one of the three questions.) Some of the new material appears at the end the essay. This material should be followed by a conclusion which summarizes the key points; the paper as it stands ends abruptly. Overall the author's revisions are minor to moderate but improve the quality of the manuscript. I would still like the author to add a couple of sentences which identify the key theoretical contributions of this paper.

Author Response

Reviewer's comments: This material should be followed by a conclusion which summarizes the key points; the paper as it stands ends abruptly. Overall the author's revisions are minor to moderate but improve the quality of the manuscript. I would still like the author to add a couple of sentences which identify the key theoretical contributions of this paper.

Response: I thank the author for rereading the paper. My response to his comments is as follows. 

I have reorganized and expanded the conclusion. The key points have been summarized in the conclusion, particularly in lines 941-47. Lines 843-850, which previously were at the end of the conclusion, have been shifted to the start of the conclusion. The ending is now gradual, not abrupt, and talks about the future, not the past.

The key theoretical contribution of the paper has been explained (lines 186-91).